# OpenReview forum: "Reversible Watermark-Guided Data Obfuscation to Prevent Exploitation by Unauthorized Models"
_ICLR.cc/2026/Conference — Submitted to ICLR 2026_

### Official Review · Reviewer_2HP9 · 2025-10-28

**Soundness:** 2
**Presentation:** 2
**Contribution:** 2
**Rating:** 4
**Confidence:** 4

**Summary:**

The paper proposes Reversible Data Obfuscation (RDO) for text: embed a reversible watermark via gradient-based search (GBS) to pick influential tokens and a constraint-guided obfuscation (CGO) rule to substitute them with top-k alternatives, so unauthorized models trained on the obfuscated corpus perform worse, while authorized users can extract the watermark and recover the original text. Claims include being the first to leverage reversible watermarking to control data usability, the two-stage GBS+CGO selection, and iterative embedding that increases capacity and progressively degrades downstream fine-tuning accuracy. Experiments on IMDB and AGNews with BERT/Qwen show sizable drops (the paper even reports sub-50% AUC/F1/accuracy in some settings), high semantic similarity by SBERT, “100%” recoverability without attack, and 92% bit-recovery under single-token tampering.

**Strengths:**

- The GBS (importance via ∂ℓ/∂e) + CGO (accept only if loss increases and label flips) is simple, reproducible, and tied to reversibility through top-2 substitution. Algorithms are explicit.

- The extraction/recovery routine is spelled out; under no attack the authors claim full recovery.

- Capacity (bpw) rises with iterations and correlates with downstream degradation; the comparison to KSPEE is a useful touchstone.

- Obfuscations crafted with (fine-tuned) DistilBERT still hurt BERT/Qwen, which suggests some transfer.

**Weaknesses:**

- Novelty positioning / missing citation. The paper repeatedly frames itself as “first” to use watermark-guided reversible hiding for usability control in text, but hiding-based unlearnable examples already exist in vision (e.g., Semantic Deep Hiding for Robust Unlearnable Examples, TIFS 2024). This related line is not cited or contrasted; the “first” claim should be scoped and toned down.
Requested point: Prior work using hiding to generate unlearnable examples in the image domain (TIFS 2024) exists and is not referenced here.

- CGO accepts edits only when they increase loss and flip the predicted label; reversibility assumes the original token is in the top-2 at both embed and extract time. Paraphrasing, back-translation, grammar/style filters, or a different tokenizer can easily move the original out of top-2, undermining both robustness and recovery. The paper evaluates only single-token substitution attacks (92% bit recognition) — much weaker than paraphrase-level transformations that real crawlers or data cleaners apply.

- Two datasets, mainly classification; no tests on summarization, QA, or instruction-tuning pipelines that re-tokenize and normalize text. Human perceptual studies are absent; imperceptibility rests solely on SBERT similarity.

- The paper cites text unlearnables (e.g., Li & Liu ’23) but doesn’t compare against strong text-native UE/poisoning baselines under the same fine-tuning protocols. The KSPEE comparison is about watermark capacity, not downstream learnability.

**Questions:**

See weakness as above.

---

> ### Author Response · Authors · 2025-11-24
>
> We appreciate your insightful feedback. We will respond to your concerns in detail.
>
> **A1**: Although both approaches are hiding-based data protection schemes, they differ significantly in their implementation techniques and application domains. The paper *Semantic Deep Hiding for Robust Unlearnable Examples* primarily focuses on protecting image data, whereas our work is centered on the protection of text data, using reversible watermarks to control the data's usage rights. To the best of our knowledge, there is no existing watermark-guided obfuscated data generation approach in the text domain.
>
> **A2**: Paraphrase-level transformations result in significant surface structure changes to the original sentence, though the semantic meaning is largely preserved. While these paraphrasing attacks do cause some loss of information, their attribution remains uncertain, and we believe they are not suitable for testing the robustness of watermark recovery. In contrast, tampering attacks, which preserve the original sentence structure and modify only a few tokens, are more appropriate for evaluating robustness and recovery, as they better retain the information.
>
> Furthermore, we extended our evaluation to multi-token substitution attacks. The results show that even with the substitution of two or three tokens, our method maintains a bit recognition rate above 85%. Specifically, two-token substitution achieves 89% bit recognition, and three-token substitution achieves 85% bit recognition.
>
> Additionally, we evaluated the obfuscation functionality after paraphrasing attacks, with detailed analysis presented in **Appendix A.4.2**. The experimental results demonstrate that, even after paraphrasing attacks, the dataset continues to maintain its obfuscation characteristics.
>
> **A3**: The CGO module of our scheme accepts edits only when they increase loss and flip the predicted label, making it unsuitable for tasks such as summarization and question answering. We plan to improve the CGO module to extend its applicability to different tasks in future work.
> For the human perceptual evaluation, we randomly sampled sentences from the dataset, marked the substituted words, and asked 10 annotators to rate the effectiveness of the watermarked sentences in maintaining the original meaning, referencing the original sentences. The score ranges from 1 to 5 (very poor to excellent). Our method achieves a score of **4**, while the unlearnable examples approach proposed by Li & Liu ’23 receives a score of **2.47**. The results demonstrate that our method outperforms in preserving the meaning of the original sentences. The detailed analysis is presented in **Appendix A.2**.
>
> **A4**: We have added a comparison between our method and unlearnable examples (Li & Liu, 2023), with a detailed analysis provided in **Appendix A.3**. Our method has a more significant impact on the model's ability to learn useful information, ultimately degrading model performance through iterative obfuscation. Additionally, our method demonstrates superior imperceptibility, emphasizing the advantages of our obfuscated data generation approach.

---

> > ### Comment · Reviewer_2HP9 · 2025-11-27
> >
> > Thanks for the detailed responses. The responses address my concerns. But I still think the originality of this paper is not convincing. If the other two reviewers stand for acceptance, I would like to raise my rating.

---

> > > ### Author Response · Authors · 2025-11-27
> > >
> > > We clarify the conceptual and technical differences between our work and prior hiding-based unlearnable examples in vision (e.g., Semantic Deep Hiding for Robust Unlearnable Examples, TIFS 2024). While both approaches fall under the general umbrella of “hiding-based data protection,” their goals, constraints, and mechanisms differ fundamentally.
> > >
> > > **1. Different problem setting and usability objective.**
> > >
> > > The goal of our work is to create watermarked obfuscated text data that degrades unauthorized model training, while authorized users can extract the embedded reversible watermark to recover the original clean text losslessly for high-quality training.
> > > In contrast, the TIFS 2024 method adds random noise during recovery and therefore cannot achieve lossless reconstruction. The reconstructed images in their pipeline deviate from the originals, which compromises data fidelity and may affect the downstream model’s effective training.
> > >
> > > **2. Reversible recovery across multiple embedding–extraction iterations.**
> > >
> > > Our reversible watermarking mechanism is stable under multiple iterations of embedding and extraction, consistently reconstructing the exact original text without error.
> > > In the image-domain method, repeated embedding and recovery steps accumulate random noise, gradually degrading the recovered images. This makes multi-stage usage (a key requirement in the scenario of multi-party data transmission and authorized reuse) infeasible, and the TIFS paper also does not analyze the fidelity of recovered images.
> > >
> > > **3. Novelty in text-domain reversible watermark-guided obfuscation.**
> > >
> > > To the best of our knowledge, no prior work on unlearnable examples in the text domain provides
> > >
> > > - reversible watermark-guided obfuscated data generation,
> > > - lossless recovery, and
> > > - explicit separation of authorized vs. unauthorized usage.
> > >
> > > Existing vision work does not address these requirements and is not directly transferable to language, given the fundamentally different discrete structure and semantic constraints of text.
> > >
> > >
> > > For these reasons, our method addresses a different task formulation, introduces new technical mechanisms that enable lossless, reversible watermark extraction and text recovery, and targets the text-domain authorization-controlled data usability problem that has not been explored previously.
> > >
> > > We will revise the manuscript to explicitly cite and contrast the TIFS 2024 work and to frame our contributions with appropriately scoped claims.

---

### Official Review · Reviewer_KdRA · 2025-10-30

**Soundness:** 3
**Presentation:** 2
**Contribution:** 3
**Rating:** 4
**Confidence:** 3

**Summary:**

This paper introduces a novel reversible watermark-guided data obfuscation (DO) method that combines gradient-based search (GBS) and constraint-guided obfuscation (CGO) to prevent unauthorized model training while allowing data recovery for authorized users. The method significantly degrades unauthorized model performance, but only uses two datasets (IMDB, AG News), which limits its generalizability. It also lacks comparison with existing techniques in data poisoning and adversarial attacks. Further comparisons and more diverse datasets are needed to validate its effectiveness and practicality.

**Strengths:**

1. This paper presents an innovative reversible watermark-guided data obfuscation (DO) method, which ensures ownership tracking and privacy protection without exposing the data content and prevents unauthorized model use.
2. The background of this paper involves data privacy protection and intellectual property protection, which is of significant practical importance in the context of the growing reliance on large-scale datasets in deep learning models, particularly in preventing data misuse.
3. The method is applicable to not only the IMDB and AG News datasets, but also performs effectively across various models such as Qwen and BERT, maintaining stable results across multiple classification tasks.

**Weaknesses:**

1. The paper only uses the IMDB and AG News datasets, limiting the generalizability of the method. While these datasets are representative, to further demonstrate the broad applicability of the method, it should be extended to more datasets, such as multilingual tasks and cross-domain datasets.
2. The paper mentions that watermark obfuscated data effectively prevents unauthorized training, but does not thoroughly explore the impact of adversarial attacks, especially after the data is tampered with or disturbed.
3. The method presented in the paper is innovative, but it lacks sufficient experimental data and analysis when compared with existing data obfuscation techniques (such as data poisoning and unlearnable samples).

**Questions:**

1. The paper only uses IMDB and AG News datasets. Is there any plan to extend the method to more datasets, especially multilingual tasks and cross-domain validation, such as tasks beyond sentiment analysis?
2. Have you considered the robustness of recovery ability across different tasks or datasets, especially when data is tampered or noisy data is involved?

---

> ### Author Response · Authors · 2025-11-24
>
> Thanks for your valuable comments. We will explain your concerns point by point.
>
> **A1**: Our current experiments focus on the IMDB and AG News datasets, which were selected to validate the effectiveness of our method. With the increasing demand for cross-lingual transfer learning and multilingual applications, extending the method to multilingual tasks is an important direction for future work. We plan to apply our method to Chinese text classification tasks.
> Regarding cross-domain validation, our method is scalable to a wide range of classification tasks. Beyond sentiment analysis, we also aim to extend the approach to more complex scenarios such as text summarization and question answering. For example, one potential direction is to generate obfuscated data by evaluating the semantic distance between model-generated answers and predefined reference answers, which would allow our framework to be adapted to summarization and QA tasks.
>
> **A2**:  We discuss the robustness of recovery ability when data is tampered in Appendix A.4.1. Experimental results show that even after tampering, our scheme achieves a bit recognition rate of 92%, demonstrating its robustness against such attacks.

---

### Official Review · Reviewer_9Dx5 · 2025-10-30

**Soundness:** 2
**Presentation:** 3
**Contribution:** 2
**Rating:** 4
**Confidence:** 3

**Summary:**

The paper addresses the problem of unauthorized data use for training commercial deep learning models. It proposes a novel method called Reversible Data Obfuscation , which aims to make data unexploitable for unauthorized parties while remaining fully recoverable and usable for authorized parties. The core of the method is a reversible watermarking scheme that embeds "model-sensitive perturbations" into text. For unauthorized users, training on this obfuscated data leads to severe performance degradation, with accuracy on classification tasks dropping below 50% after iterative embedding. For authorized users, who possess the key, they can perfectly reverse the process, extract the watermark, and recover the original high-quality data for effective training.

**Strengths:**

- The paper's primary strength is its novel problem formulation and solution. It bridges the gap between two previously separate fields: data poisoning/unlearnable examples and watermarking. The idea of a conditionally destructive dataset (unlearnable for unauthorized users but recoverable for authorized ones) is a original and valuable contribution.
- The paper is exceptionally well-written and easy to follow. The high-level concept is perfectly illustrated in Figure 1, and the complex embedding mechanism is demystified in Figure 2. The methodology is further clarified with well-defined algorithms.
- This work provides a practical, "lock-and-key" tool for data owners (e.g., artists, writers, researchers) to protect their public data from unauthorized scraping and exploitation by large model developers. This moves beyond passive protection (ownership verification) and destructive protection (unlearnable examples) to a more flexible and powerful form of data control.

**Weaknesses:**

- The extraction process requires the authorized user to have the set of embedded positions. This "key" is a significant secret. The paper does not discuss how the set of embedded positions is managed, stored, or securely shared. If the set of embedded positions is simply "the list of all indices that were successfully modified," an attacker with knowledge of the method might try to identify these locations (e.g., by finding tokens with low probability under a standard LM). This aspect of the threat model needs to be elaborated.
- The robustness analysis in Appendix A.2 is a good start but is limited to a very weak attack: "substitutes one token per sample". A more realistic and powerful attack would be for an unauthorized user to paraphrase the entire dataset using a powerful LLM. This would almost certainly destroy the fragile token-level candidate relationships at the embedded positions, rendering extraction impossible and likely mitigating the obfuscation's effect. This is a major, unaddressed threat.

**Questions:**

I have the following questions for the authors, which would help clarify the limitations and practical applicability of the work:
- The extraction algorithm requires the set of embedded positions. How do you propose this "key" is managed in a real-world scenario? Is it a static list, or is it (or its starting seed) a secret shared between the owner and authorized users? If it's a static list, what prevents an attacker from trying to discover these positions?
- A key threat not addressed is a paraphrasing attack, where an unauthorized user re-writes the dataset with an LLM before training. Have you tested the method's robustness against this? Would this not break both the watermark recovery and the obfuscation effect?
- Could you please provide qualitative examples (like those in Table 3 ) of text samples after 5, 7, and 9 iterations of embedding? I am interested to see if the text remains "imperceptible to humans"  after so many cumulative modifications.

---

> ### Author Response · Authors · 2025-11-24
>
> Thank you for your valuable feedback. We have made revisions based on your suggestions.
>
> A1: The key can be managed by a trusted third party, with authorized users granted access to it. It is a customized list tailored to the data owner's requirements, meaning different data and user watermarks will generate distinct lists.
>
> Our method selects the tokens to be modified based on watermark bits, choosing from the top-4 predictions. We draw the distribution of the original token among the top-4 predictions before modification, as well as the token distribution in the modified text after watermark embedding. As seen, the token distribution does not exhibit significant changes, so an attacker cannot easily identify these positions without knowledge of the "list of successfully modified indices." Additionally, even if the token appears in the top-4 predictions, the attacker cannot confirm whether watermark embedding operation is performed at that position to generate obfuscated data. The token itself may locate in the top-3 or top-4 of the prediction results (a relatively low-probability label), and no watermark operation has been applied. Therefore, the "list of successfully modified indices" is essential, as it is the core element ensuring the correct recovery of the data.
>
> A2: Paraphrasing attacks rewrite original sentences with LLM such that surface structure changes significantly while semantic meaning is largely preserved. We tested our method’s robustness against this type of attack, and the obfuscation effect is remained. Since paraphrasing alters the sentence structure, it disrupts the extraction of watermark information. More importantly, the paraphrasing process itself inevitably removes or distorts certain information, as discussed in **Appendix A.4.2**. As a result, the post-paraphrasing textual attribution becomes uncertain, and thus we do not further analyze the watermark's robustness or recoverability in this context.
>
> In Appendix A.4.2, we provide additional analysis and present the performance of our method after paraphrasing attacks. The results show that even after paraphrasing, the dataset still retains its obfuscation characteristics.
>
> A3: We provide examples in Table R1.
>
> ### Table R1: Some examples of text samples after 5, 7, and 9 iterations of embedding.
>
> | **Obfuscation Iteration** | **Data** |
> |---------------------------|----------|
> | **0** | “( cnn ) -- a longtime **_associate_** of al qaeda leader osama bin laden surrendered to saudi arabian officials tuesday, a saudi interior **_ministry_** official said.” “but it is unclear what role, if any, **_khaled_** al-harbi may have had in any terror attacks because no public charges have been **_filed_** against him.” “the saudi government — in a statement released by its embassy in washington — called al-harbi's surrender ‘the latest direct **_result_**’ of its limited, **_one_**-month offer of leniency to terror suspects.” this is great! i **_hope_** this **_really_** starts to pay off. creative **_solutions_** to terrorism that don't involve violence. |
> | **5** | “( cnn ) -- a longtime **_friend_** of al qaeda leader osama bin laden surrendered to saudi officials tuesday, a saudi interior **_minister_** official said.” “but it is unclear what role, if any, khaled al-harbi may have had in any terror attacks because no public charges have been **_leveled_** against him.” “the saudi government — in a statement released by its embassy in washington — called al-harbi's surrender ‘the latest direct result’ of its limited, **_12_**-month offer of leniency to terror suspects.” this is great! i hope this really starts to pay off. creative **_approach_** to terrorism that don't involve violence. |
> | **7** | “( cnn ) -- a longtime friend of al qaeda leader osama bin laden surrendered to saudi officials tuesday, a saudi interior minister official said.” “but it is unclear what role, if any, khaled al-harbi may have had in any terror attacks because no public charges have been leveled against him.” “the saudi government — in a statement released by its embassy in washington — called al-harbi's surrender ‘the latest direct result’ of its limited, 12-month offer of leniency to terror suspects.” this is great! i **_believe_** this **_situation_** starts to pay off. creative approach to terrorism that don't involve violence. |
> | **9** | “( cnn ) -- a longtime friend of al qaeda leader osama bin laden surrendered to saudi officials tuesday, a saudi interior minister official said.” “but it is unclear what role, if any, **_ghaled_** al-harbi may have had in any terror attacks because no public charges have been leveled against him.” “the saudi government — in a statement released by its embassy in washington — called al-harbi's surrender ‘the latest direct **_outcome_**’ of its limited, 12-month offer of leniency to terror suspects.” this is great! i believe this situation starts to pay off. creative approach to terrorism that don't involve violence. |

---

> ### Author Response · Authors · 2025-11-27
> **Follow-up on Rebuttal Submission for ICLR Paper [Paper ID: 10114]**
>
> Dear AC, SAC, PC and Reviewers,
>
> I hope you're doing well. I am reaching out to follow up on the rebuttal submission for our ICLR paper titled "Reversible Watermark-Guided Data Obfuscation to Prevent Exploitation by Unauthorized Models" (Paper ID: 10114). We have carefully considered the reviewers' comments and have conducted additional experiments to address the feedback. The updated results have been included in the rebuttal submission.
>
> As the rebuttal phase is coming to a close, we would be grateful for any updates on the status of our submission. We understand that the review process takes time, and we are happy to provide any further clarifications or additional information if necessary.
>
> Thank you for your time and consideration. We look forward to your response.
>
> Best regards

---

### Meta-Review · Area_Chair_cVzU · 2026-01-07

**Summary:**

The paper presents "Reversible Data Obfuscation", the (claimed) first method that leverages watermark in a reversible manner to control data usability for target models. The topic is timely in trustworthy AI, that being said, overall the reviewers remained unenthusiastic to this paper and pointed out several issues that the authors are encouraged to incorporate in next iterations of their paper, especially some undressed ones such as those pointed out by 2HP9.

**Reviewer Concerns:**

most minor concerns were resolved by the authors
reviewer 2HP9 concern on comparing with the use of hiding to generate unlearnable examples seems to be unresolved to me.

**Reviewer Scores:**

not sure how to reply in the case of this paper, because the reviewers did not answer the authors rebuttal including before the openreview issues

---

### Decision · Program_Chairs · 2026-01-26

Reject